# Bootstrap-quantile ridge estimator for linear regression with applications

**Irum Sajjad Dar, Sohail Chand** ⓘ *

College of Statistical Sciences, University of the Punjab, Lahore, Pakistan

* sohail.stat@pu.edu.pk

## Abstract

Bootstrap is a simple, yet powerful method of estimation based on the concept of random sampling with replacement. The ridge regression using a biasing parameter has become a viable alternative to the ordinary least square regression model for the analysis of data where predictors are collinear. This paper develops a nonparametric bootstrap-quantile approach for the estimation of ridge parameter in the linear regression model. The proposed method is illustrated using some popular and widely used ridge estimators, but this idea can be extended to any ridge estimator. Monte Carlo simulations are carried out to compare the performance of the proposed estimators with their baseline counterparts. It is demonstrated empirically that *MSE* obtained from our suggested bootstrap-quantile approach are substantially smaller than their baseline estimators especially when collinearity is high. Application to real data sets reveals the suitability of the idea.

## 1. Introduction

Multiple Linear regression (*MLR*) is one of the most popular tool used by practitioners due to its simplicity and attractive properties. However, it has been observed that many complex and interesting situations may exist. For example, collinearity among predictors is an imperative problem [1] faced by many researchers. In this situation, various methods have been developed to cope with collinearity problem. Among such methods are Ridge regression [2], Principal component regression [3], Partial least squares regression [4], and Continuum regression [5]. Many researchers have used such methods in accelerated failure time models (see e.g. [6–8]).

However, ridge regression (*RR*) is the most popular and is extensively used in practice due to its relatively low computational cost, strong theoretical guarantees, and interpretability [9]. It can be a useful technique for estimating the coefficients of the model as it provides precise estimates by introducing some bias in the regression model. This shrinkage method allows all the considered covariates to be included in the model but with shrunken coefficients.

Consider the classic *MLR* model:

$$y = X\beta + \varepsilon, \tag{1}$$

where $y_{(n \times 1)}$ is the vector of the dependent variable, $X_{(n \times p)}$ is the design matrix, $\beta_{(p \times 1)}$ is a vector of unknown regression parameters i.e. $\beta = (\beta_0, \beta_1, \ldots, \beta_p)'$, without loss of generality, we assume $\beta_0 = 0$, $\varepsilon_{(n \times 1)}$ is a vector of error random variables with mean vector 0 and variance-covariance matrix $\sigma^2 I_n$, where $n$ denotes the number of observations, $p$ represents the number

**Data Availability Statement:** Published data has been used. A complete reference is provided in the manuscript.

**Funding:** The authors received no specific funding for this work.

**Competing interests:** The authors have declared that no competing interests exist.

of predictors and $I_n$ is the identity matrix of order n. The ordinary least square (*OLS*) estimator of $\beta$ is given as:

$$\hat{\beta}_{OLS} = (X'X)^{-1}X'y.$$

which will mainly depend on the characteristics of the matrix $X'X$. When the predictors are correlated i.e. (at least one of the predictors is a linear combination of other predictors), the matrix $X'X$ becomes ill-conditioned. It implies that for some of the eigen values of $X'X$ becomes close to zero. Thus, the variance of the *OLS* estimator is inflated so that one cannot obtain stable estimates [10].

To circumvent this situation, [2] introduced the concept of *RR* in which a non-negative constant $k$ is added to the diagonal elements of $X'X$ so that the estimates or stable estimates can be obtained at the cost of some bias.

The *RR* estimator is defined as follows:

$$\hat{\beta}_{RR} = (X'X + kI_p)^{-1}X'y, \qquad k > 0,$$

where $k$ is known as the ridge parameter or the biasing parameter. The *RR* has become an accepted alternative to *OLS* for the ill-conditioned design matrix, but there is no theoretical agreement on the optimal choice of $k$. The selection of $k$ is yet one of the most challenging and fundamentally exciting existing problem. The main concern is to find the value of $k$ such that the reduction in the variance term is greater than the increase in the squared bias. A striking diversity persists in choices for $k$ as advocated in the literature (see e.g. [2, 11–21]) among others and references therein; but to search for the optimal value of the ridge parameter remains open. Also, to mitigate the effect of multicollinearity in different situations many researchers have employed *RR* in Beta regression [22], Gaussian linear model [23], Logistic regression [24], Poisson regression [25], Tobit regression [26], to mention but a few.

Bootstrap initiated by [27] is a generic statistical method for assessing the accuracy of an estimator. The core mechanism is based on the concept of random sampling with replacement. Many bootstrap techniques have been developed to address a large variety of statistical problems (see e.g. [28–32]). The common use of bootstrapping is the construction of confidence intervals, the approximation of critical points in hypothesis tests, calculating standard errors and bias, etc. However, we will show that the performance of any existing ridge estimator can be improved by replacing it with some appropriate quantile of its bootstrap distribution. The basic component in bootstrap methods is to replace a true distribution function with its empirical estimator. A very primitive understanding of bootstrap methods is that it does not rely on any distributional assumptions such as normality and can estimate the standard error of any complicated estimator without any theoretical calculations [29]. Such a method ensures that the statistics evaluated are accurate and unbiased as much as possible.

The work in this manuscript mainly came out of the combination of the ideas from [33], as they mentioned that the bootstrap approach for the selection of ridge parameter is justified as it is based on repeated and independent estimates of multiple predictions and [18] who used to calculate the γth quantile of the ridge estimator originally proposed by [2] presented in Eq (4).

We combine and extend their ideas to improve the performance of already existing ridge estimators while maintaining the same general idea. The objective of this manuscript is to propose a novel bootstrap-quantile ridge (*BQR*) estimator, and its computational algorithm and also to evaluate and compare its performance with the baseline estimator. That is based on the approach to arrive at an optimal choice of ridge parameter that will yield mean square error *(MSE)* substantially smaller as compared to its baseline counterpart. The paper would also contribute to the existing literature by offering new insights into the selection of ridge parameter.

The remainder of this article is structured as follows: Statistical methodology along with a brief review of some popular and widely used existing ridge estimators and our suggested *BQR* versions of these estimators are described in Sec. 2. A Monte Carlo simulation study has been conducted and its results are provided and discussed in Sec. 3. A real-life application is provided in Sec. 4. Finally, Sec. 5 concludes the article.

## 2. Statistical methodology

Considering the model given in (1), suppose that $U$ is ($p$ x $p$) orthogonal matrix such that $U'U = I_p$ and $U'X'XU = \Lambda$ where $\Lambda = \text{diag}(\lambda_1, \lambda_2, \ldots, \lambda_p)$ contains the eigen values of the matrix $X'X$. Then, the canonical form of model (1) is expressed as y = $Z\alpha + \varepsilon$, where $Z = XU$ and $\alpha = (\alpha_1, \alpha_2, \alpha_3, \ldots, \alpha_p)' = U'\beta$.

The *RR* estimator in the canonical form is given as: $\hat{\alpha}(k) = (Z'Z + K)^{-1}Z'Y$, where $K = diag(k_1, k_2, \ldots, k_p), k_w > 0 \text{ for } w = 1, 2, \ldots, p$. The *OLS* estimate of $\alpha$ is $\hat{\alpha} = \Lambda^{-1}Z'y$. The *MSE* of $\hat{\alpha}(k)$ is given as follows:

$$MSE(\hat{\alpha}(k)) = \hat{\sigma}^2 \sum_{w=1}^{p} \frac{\lambda_w}{(\lambda_w + k_w)^2} + \sum_{w=1}^{p} \frac{k_w^2 \alpha_w^2}{(\lambda_w + k_w)^2}. \tag{2}$$

where, $\hat{\sigma}^2$ represents the error variance of the model (1), $\alpha_w$ is the $w^{th}$ value of $\alpha$ and $\lambda_w$ is the $w^{th}$ eigen value of the matrix $X'X$.

### 2.1. Existing estimators

This section discusses some of the popular existing estimators while our suggested *BQR* versions are provided in a subsequent section. [2] proposed ridge estimator as an alternative to the *OLS* estimator for use in the presence of collinearity among predictors. They suggested $k_w$ as the ratio of estimated error variance ($\hat{\sigma}^2$) and $w^{th}$ estimate of $\alpha$ using *OLS* as follows:

$$\hat{k}_w = \frac{\hat{\sigma}^2}{\hat{\alpha}_w^2}, \qquad w = 1, 2, \ldots, p. \tag{3}$$

where $\hat{\sigma}^2 = \sum_{j=1}^{n} \hat{\varepsilon}_j / (n - p)$ represents the residual mean square estimate which is an unbiased estimator of $\sigma^2$ and $\hat{\alpha}_w$ is the $w^{th}$ element of $\hat{\alpha}$ which is an unbiased estimate of α.

Furthermore, they found that the best strategy for achieving an optimal estimate is to determine a single value for $p$ ridge parameters i.e. to replace $k_w$ with $k$ for all $w$ defined in Eq (3) and hence suggested the following estimator:

$$\hat{k}_{HK} = \frac{\hat{\sigma}^2}{\hat{\alpha}_{max}^2}, \tag{4}$$

where $\hat{\alpha}_{max} = \max(\hat{\alpha}_1, \hat{\alpha}_2, \hat{\alpha}_3, \ldots, \hat{\alpha}_p)$.

Later, [11] proposed a ridge estimator based on the harmonic mean of $\hat{k}_w$; harmonic mean is used as a way to prevent the small $\alpha_w$, that have little predicting power and defined as

$$\hat{k}_{HKB} = \frac{p\hat{\sigma}^2}{\sum_{w=1}^{p} \hat{\alpha}_w^2}, \tag{5}$$

[12] suggested the following ridge estimator by using weights of eigen values

$$\hat{k}_{HSL} = \hat{\sigma}^2 \frac{\sum_{w=1}^{p} (\lambda_w \hat{\alpha}_w)^2}{(\sum_{w=1}^{p} \lambda_w \hat{\alpha}_w^2)^2}, \tag{6}$$

where $\lambda_w$ is the $w^{th}$ eigen value.

[14] modified $\hat{k}_w$ ridge estimator by taking its arithmetic mean and geometric mean as follows:

$$\hat{k}_{AM} = \frac{1}{p} \sum_{w=1}^{p} \frac{\hat{\sigma}^2}{\hat{\alpha}_w^2}, \tag{7}$$

$$\hat{k}_{GM} = \hat{\sigma}^2 / (\prod_{w=1}^{p} \hat{\alpha}_w^2)^{1/p}. \tag{8}$$

For the sake of simplicity, in this paper, we will name these estimators as *HK*, *HKB*, *HSL*, *AM*, and *GM* defined in Eqs (4)–(8).

## 2.2. Novel bootstrap-quantile estimators

Consider any ridge estimator $\hat{k}$ and its estimates obtained for $B$ bootstrap samples $\hat{k}^{*(u)}$; $u = 1, 2, 3, \ldots, B$. The estimates are ordered in magnitude as $\hat{k}^{*(1)} \leq \hat{k}^{*(2)} \leq \hat{k}^{*(3)} \leq \ldots \leq \hat{k}^{*(B)}$. Let $\{\hat{k}_\gamma^*, 0 < \gamma < 1\}$ be the 100 $\gamma$ th quantile of $\{\hat{k}^{*(1)}, \hat{k}^{*(2)}, \hat{k}^{*(3)}, \ldots, \hat{k}^{*(B)}\}$ then the proposed *BQR* is:

$\hat{k}_\gamma^* = \{\hat{k}^{*(1)}, \hat{k}^{*(2)}, \hat{k}^{*(3)}, \ldots, \hat{k}^{*(B)}\}_\gamma$ such that $P(\hat{k}^{*(u)} < \hat{k}_\gamma^*) = \gamma$.

It can be easily noted that the performance of any ridge estimator can be improved by an appropriate selection of quantile level. Moreover, it has been noticed that generally upper quantile levels are the appropriate choice in this regard. Thus, we have a *BQR* version $\hat{k}_\gamma^*$ of any ridge estimator $\hat{k}$ such that the suggested *BQR* version yields the smallest *MSE*.

As in Eq (2), the *MSE* of the ridge in the canonical form is mentioned, therefore

$$MSE(\hat{\alpha}(\hat{k}^{*(u)})) = \sigma^2 \sum_{w=1}^{p} \frac{\lambda_w}{(\lambda_w + \hat{k}_w^{*(u)})^2} + \sum_{w=1}^{p} \frac{\hat{k}_w^{*(u)} \alpha_w^2}{(\lambda_w + \hat{k}_w^{*(u)})^2},$$

There exists a real number $\hat{k}_\gamma^*$ in the interval; $\min(\hat{k}^{*(u)}) < \hat{k}_\gamma^* < \max(\hat{k}^{*(u)})$, such that

$$MSE(\hat{k}) - MSE(\hat{k}_\gamma^*) > 0 \ i.e. \ MSE(\hat{k}_\gamma^*) < MSE(\hat{k}).$$

**2.2.1 Theoretical justification of the proposed method.** Let *G* be the cumulative density function (CDF) of the estimator of ridge estimator *k*. The $\gamma th$ quantile of the distribution can be defined as $P(k_\gamma) = \gamma$, such that $\hat{k}_\gamma(G) = G^{-1}(\gamma)$. Let $\{(X_j^*, y_j^*) : j = 1, 2, \ldots, n\}$ be a bootstrap sample then selecting "B" bootstrap samples and estimating the ridge parameter i.e. $\hat{k}^{*(1)}, \hat{k}^{*(2)}, \hat{k}^{*(3)}, \ldots, \hat{k}^{*(B)}$. We can define the empirical distribution function

$$\hat{G}_n^*(k) = \frac{1}{B} \sum_{u=1}^{B} I(\hat{k}^* \leq k).$$

It is important to note that the target function, $\hat{k}_\gamma(G)$, is not linear. Let $\theta_k$ be a point mass at location $k$, then we can define the statistical function, $T_k$, and influence function see e.g. [29] as follows:

$$L_G(k) = \lim_{\delta \to 0} \frac{T_k[(1-\delta)G + \delta\theta_k] - T_k(G)}{\delta}.$$

Under the condition, $T_k$ is smooth then $L_F(k)$ is also smooth, which leads to the following

$$V_k(G) = \int L_G^2(k)dG(k)$$

The bootstrap is simply a plug-in estimate i.e. $V_k(\hat{G}_n) = \int L_{\hat{G}_n}^2(k)d\hat{G}_n(k)$ and

$$V_k(\hat{G}_n) \approx V_k(G).$$

This consequently leads to the validity of bootstrap consistency.
The influence function of $T_{k_\gamma}$ is

$$L_G(k) = \frac{\gamma}{P(G^{-1}(\gamma))}.$$

Where $P$ is the pdf of $G$. Thus, using the delta method see. e.g. [34]

$$\sqrt{n}\left(T_{k_\gamma}(\hat{G}_n) - T_{k_\gamma}(G)\right) \to N\left(0, \frac{\gamma^2}{p^2(G^{-1}(\gamma))}\right).$$

This derives the asymptotic distribution of the bootstrap quantile estimator and also the variance of the estimator.

**2.2.2. Computation of BQR.** The proposed *BQR* estimators are computed using the methodology described in Algorithm-1.

```
Algorithm-1: Bootstrap-Quantile Estimator
1. For each j = 1,2,···n, generate random samples {(X_j,y_j): j = 1,2,···,
n} under the specified DGP.
2. Select B bootstrap from {(X_j,y_j):j = 1,2,···,n} i.e.
{(X_j^{*(u)},y_j^{*(u)}):j = 1,2,...,n,u = 1,2,...,B}
3. For each bootstrap sample compute k^{*(u)}:u = 1,2,···,B.
4. Calculate the γ^{th} quantile of k̂^{*(u)} such that
P(k̂^{*(u)} ≤ k̂_γ^*) = γ and MSE(k̂_γ^*) < MSE(k̂).
```

The proposed *BQR* estimators corresponding to considered ridge estimators are denoted by $HK_\gamma^*$, $HKB_\gamma^*$, $HSL_\gamma^*$, $AM_\gamma^*$, and $GM_\gamma^*$.

## 2.3. Performance evaluation criteria

The *MSE* criterion is used to evaluate the performance of our proposed estimators with their baseline counterparts. Previously this evaluation criterion has been used in numerous researches such as [14, 18, 35, 36] among many others.

The *MSE* is given as:

$$MSE(\hat{\alpha}) = E(\hat{\alpha} - \alpha)'(\hat{\alpha} - \alpha).$$

Since the theoretical comparison of the ridge estimators presented in Eqs (4)–(8) with their *BQR* version is not possible. So, we establish empirically and for this purpose, Monte Carlo simulations are used to assess the performance of the considered and proposed estimators. To

further quantify the strength of the proposed estimators over baseline estimators, the improved percentage indicator of *MSE* with respect to *BQR* is calculated as

$$P_{MSE} = \frac{(MSE - MSE_{\hat{\gamma}}^*)}{MSE} \times 100.$$

where $P_{MSE}$ indicates the percentage of increase/decrease in *MSE* due to the *BQR* estimator in comparison with their baseline counterpart. Thus, theoretically, a positive $P_{MSE}$ indicates improvement achieved while a negative percentage indicates deterioration due to the use of *BQR* compared to their baseline estimators.

## 3. Numerical evaluation

In this section, we will briefly describe the data generation process (*DGP*) together with the important factors like sample size ($n$), error variance ($\sigma^2$), dimension ($p$), collinearity level ($\rho$), and error term distribution which are varied in the simulation study to see the behavior in different settings. The results of a simulation study for baseline and their corresponding *BQR* versions are also presented.

### 3.1 Simulation design

Keeping in view the previous studies, we take the *DGP* as mentioned by [14, 20, 21, 37, 38] and is mentioned below.

$$x_{jw} = (1 - \rho^2)^{1/2} z_{jw} + \rho z_{jp}, \quad w = 1, 2, \ldots, p. j = 1, 2, \ldots, n.$$

here $z_{jw}$ are the independent pseudo-random numbers drawn from a standard normal distribution and $\rho$ is the degree of correlation between any two predictors. Further, the dependent variable, $y$ is generated as:

$$y_j = \beta_0 + \beta_1 x_{j1} + \beta_2 x_{j2} + \ldots + \beta_p x_{jp} + \varepsilon_j, \quad j = 1, 2, \ldots, n.$$

where $\varepsilon_j$ is the random error generated from a normal distribution with zero mean and variance $\sigma^2$ Here, without loss of generality, we can assume $\beta_0 = 0$ i.e. the intercept of the regression model is zero. The experiment is replicated *M* times and the *MSE* of the estimators is computed using the following formula:

$$MSE(\hat{\alpha}_k) = \frac{1}{M} \sum_{v=1}^{M} \sum_{w}^{p} (\hat{\alpha}_{vw} - \alpha_w)^2, \quad v = 1, 2, \ldots, M, w = 1, 2, 3, \ldots, p,$$

where $\hat{\alpha}_{vw}$ represents one of the above mentioned estimates of $w^{th}$ true regression parameter $\alpha_w$ in the $v^{th}$ replication.

The design matrix $X$ is generated to investigate the effects of four levels of collinearity, i.e., $\rho$ = 0.70, 0.80, 0.90, and 0.99. These values will cover a wide range of low, moderate, and strong correlations among the variables. Three levels of error variance ($\sigma^2$ = 0.5, 1, 5). Whereas $n$ and $p$ are varied proportionally i.e. we consider the following cases.

**Case 1:** $n$ = 25, $p$ = 4, $\sigma^2$ = 0.5, 1, 5 and $\rho$ = 0.70, 0.80, 0.90 and 0.99.

**Case 2:** $n$ = 50, $p$ = 8, $\sigma^2$ = 0.5, 1, 5 and $\rho$ = 0.70, 0.80, 0.90 and 0.99.

**Case 3:** $n$ = 100, $p$ = 10, $\sigma^2$ = 0.5, 1, 5 and $\rho$ = 0.70, 0.80, 0.90 and 0.99.

Further to explore the behavior of the proposed *BQR* for large sample size, as an illustration, we have considered,

**Case 4:** $n$ = 200, 03c3 $\sigma^2$ = 1, $\rho$ = 0.99 with varying choices of number of predictors i.e., $p$ = 4, 8, 10, 16, 32.

**Table 1. Estimated *MSE* when the distribution of error term is $N(0,\sigma^2)$ with $n=25$, $p=4$.**

| $\sigma^2$ | OLS | HK | $HK_\gamma^*$ | HKB | $HKB_\gamma^*$ | HSL | $HSL_\gamma^*$ | AM | $AM_\gamma^*$ | GM | $GM_\gamma^*$ |
|---|---|---|---|---|---|---|---|---|---|---|---|
| \multicolumn{12}{c}{$\rho = 0.7$} |
| 0.5 | 0.105 | 0.097 | **0.064** | 0.082 | **0.051** | 0.097 | **0.079** | 0.305 | **0.045** | 0.080 | **0.046** |
| 1 | 0.401 | 0.342 | **0.172** | 0.255 | **0.168** | 0.351 | **0.277** | 0.397 | **0.145** | 0.185 | **0.127** |
| 5 | 11.232 | 5.024 | **1.168** | 4.341 | **1.763** | 4.797 | **2.502** | 1.084 | **0.661** | 1.783 | **0.604** |
| \multicolumn{12}{c}{$\rho = 0.8$} |
| 0.5 | 0.301 | 0.206 | **0.074** | 0.151 | **0.067** | 0.206 | **0.123** | 0.299 | **0.045** | 0.075 | **0.042** |
| 1 | 1.081 | 0.665 | **0.203** | 0.465 | **0.199** | 0.510 | **0.294** | 0.372 | **0.125** | 0.195 | **0.154** |
| 5 | 20.711 | 9.634 | **1.514** | 8.811 | **3.156** | 6.484 | **2.509** | 1.104 | **0.559** | 2.040 | **0.552** |
| \multicolumn{12}{c}{$\rho = 0.9$} |
| 0.5 | 0.398 | 0.259 | **0.081** | 0.190 | **0.074** | 0.260 | **0.146** | 0.187 | **0.052** | 0.137 | **0.035** |
| 1 | 1.138 | 0.793 | **0.422** | 0.585 | **0.228** | 0.631 | **0.324** | 0.296 | **0.115** | 0.317 | **0.094** |
| 5 | 35.32 | 17.89 | **2.497** | 15.110 | **5.038** | 6.771 | **2.277** | 1.285 | **0.525** | 2.534 | **0.585** |
| \multicolumn{12}{c}{$\rho = 0.99$} |
| 0.5 | 2.909 | 0.899 | **0.116** | 0.701 | **0.137** | 0.614 | **0.137** | 0.327 | **0.037** | 0.483 | **0.035** |
| 1 | 11.88 | 6.533 | **0.375** | 5.601 | **1.446** | 0.592 | **0.106** | 0.794 | **0.083** | 1.465 | **0.179** |
| 5 | 190.3 | 45.27 | **4.221** | 39.651 | **14.20** | 8.097 | **1.153** | 1.225 | **0.466** | 10.06 | **1.123** |

In this study, the number of bootstrap samples is taken to be 200 (i.e. $B = 200$). The estimated *MSE* on different combinations of $n$, $p$, $\sigma^2$, and $\rho$, when the error term is normally distributed is presented in Tables 1–3. Furthermore, to study the effect of the non-normal error term, we have generated the error terms from $t$-distribution with 2 degrees of freedom i.e. $\varepsilon_j \sim t(2)$ and also from $F$-distribution with 4 and 16 degrees of freedom i.e. $\varepsilon_j \sim F(4,16)$. The results of the estimated *MSE* of the classical ridge and our proposed *BQR* estimators when the error term is generated from the $t$ and $F$ distribution are presented in Tables 4 and 5 respectively. We used the R programming language version 4.1.0 to perform all the calculations that were made for the analysis of the estimators.

**Table 2. Estimated *MSE* when the distribution of error term is $N(0,\sigma^2)$ with $n = 50$, $p = 8$.**

| $\sigma^2$ | OLS | HK | $HK_\gamma^*$ | HKB | $HKB_\gamma^*$ | HSL | $HSL_\gamma^*$ | AM | $AM_\gamma^*$ | GM | $GM_\gamma^*$ |
|---|---|---|---|---|---|---|---|---|---|---|---|
| \multicolumn{12}{c}{$\rho = 0.7$} |
| 0.5 | 0.242 | 0.111 | **0.037** | 0.086 | **0.061** | 0.111 | **0.102** | 0.109 | **0.030** | 0.052 | **0.030** |
| 1 | 0.675 | 0.455 | **0.385** | 0.270 | **0.210** | 0.458 | **0.413** | 0.148 | **0.080** | 0.111 | **0.078** |
| 5 | 10.37 | 5.780 | **2.182** | 3.485 | **2.417** | 4.862 | **4.335** | 0.702 | **0.411** | 1.139 | **0.492** |
| \multicolumn{12}{c}{$\rho = 0.8$} |
| 0.5 | 0.489 | 0.258 | **0.169** | 0.162 | **0.082** | 0.258 | **0.193** | 0.121 | **0.023** | 0.043 | **0.026** |
| 1 | 0.948 | 0.579 | **0.416** | 0.313 | **0.214** | 0.556 | **0.450** | 0.208 | **0.071** | 0.108 | **0.065** |
| 5 | 21.32 | 9.492 | **2.709** | 5.478 | **3.643** | 5.088 | **4.354** | 0.825 | **0.309** | 1.813 | **0.649** |
| \multicolumn{12}{c}{$\rho = 0.9$} |
| 0.5 | 0.520 | 0.282 | **0.168** | 0.168 | **0.083** | 0.282 | **0.196** | 0.205 | **0.025** | 0.053 | **0.021** |
| 1 | 1.509 | 0.869 | **0.595** | 0.480 | **0.322** | 0.836 | **0.654** | 0.278 | **0.048** | 0.159 | **0.058** |
| 5 | 57.24 | 14.531 | **3.375** | 9.063 | **5.337** | 4.920 | **3.923** | 0.795 | **0.277** | 1.982 | **0.672** |
| \multicolumn{12}{c}{$\rho = 0.99$} |
| 0.5 | 6.277 | 1.729 | **0.449** | 0.996 | **0.276** | 1.138 | **0.376** | 0.261 | **0.013** | 0.312 | **0.036** |
| 1 | 17.46 | 6.889 | **1.669** | 4.124 | **2.082** | 1.702 | **0.767** | 0.373 | **0.024** | 1.033 | **0.258** |
| 5 | 531.7 | 190.3 | **35.60** | 111.93 | **68.16** | 1.755 | **0.957** | 2.562 | **0.139** | 26.260 | **7.637** |

**Table 3. Estimated *MSE* when the distribution of error term is $N(0,\sigma^2)$ with $n = 100, p = 10$.**

| $\sigma^2$ | OLS | HK | $HK_\gamma^*$ | HKB | $HKB_\gamma^*$ | HSL | $HSL_\gamma^*$ | AM | $AM_\gamma^*$ | GM | $GM_\gamma^*$ |
|---|---|---|---|---|---|---|---|---|---|---|---|
| | | | | | $\rho = 0.7$ | | | | | | |
| 0.5 | 0.087 | 0.077 | **0.0741** | 0.061 | **0.051** | 0.078 | **0.075** | 0.065 | **0.021** | 0.034 | **0.019** |
| 1 | 0.352 | 0.310 | **0.2890** | 0.178 | **0.154** | 0.311 | **0.295** | 0.112 | **0.048** | 0.062 | **0.047** |
| 5 | 7.502 | 4.200 | **2.304** | 2.438 | **1.873** | 3.483 | **3.258** | 0.583 | **0.262** | 0.661 | **0.339** |
| | | | | | $\rho = 0.8$ | | | | | | |
| 0.5 | 0.132 | 0.101 | **0.094** | 0.073 | **0.075** | 0.101 | **0.095** | 0.119 | **0.098** | 0.024 | **0.016** |
| 1 | 0.502 | 0.440 | **0.391** | 0.241 | **0.202** | 0.439 | **0.406** | 0.312 | **0.039** | 0.062 | **0.039** |
| 5 | 15.79 | 5.623 | **2.672** | 3.173 | **2.382** | 3.651 | **3.369** | 0.537 | **0.229** | 0.902 | **0.393** |
| | | | | | $\rho = 0.9$ | | | | | | |
| 0.5 | 0.249 | 0.169 | **0.130** | 0.107 | **0.074** | 0.169 | **0.149** | 0.207 | **0.113** | 0.029 | **0.013** |
| 1 | 1.088 | 0.591 | **0.479** | 0.303 | **0.240** | 0.573 | **0.512** | 0.254 | **0.035** | 0.078 | **0.036** |
| 5 | 31.38 | 10.726 | **3.901** | 6.056 | **4.255** | 3.656 | **3.158** | 0.578 | **0.183** | 1.505 | **0.626** |
| | | | | | $\rho = 0.99$ | | | | | | |
| 0.5 | 2.395 | 1.326 | **0.642** | 0.685 | **0.320** | 1.066 | **0.617** | 0.111 | **0.007** | 0.179 | **0.038** |
| 1 | 9.973 | 4.899 | **1.798** | 2.581 | **1.513** | 1.930 | **1.340** | 0.207 | **0.012** | 0.615 | **0.220** |
| 5 | 240.11 | 101.02 | **29.85** | 51.94 | **34.36** | 1.525 | **1.138** | 1.811 | **0.117** | 10.015 | **3.927** |

## 3.2. Results and discussion

In this section, we have used extensive simulations under various considered scenarios. The performance of the proposed estimators and the existing estimators are assessed on the basis of the *MSE* criterion. As the *MSE* is affected by error variance, distribution of error term, dimensionality, predictor's correlation matrix, and the sample size, we have considered various combinations of these factors already discussed in the previous section. In this study, we have used 1000 Monte Carlo runs and 200 bootstrap samples and the results of *MSE* are presented in Tables 1–5. Whereas for a clearer picture Figs 1–3 exhibit percentage reduction in *MSE* ($P_{MSE}$) due to the proposed *BQR* method as compared to the classical counterpart when the distribution of error term is $N(0,\sigma^2)$, whereas Figs 4, and 5 indicates $P_{MSE}$ when the distribution of error term is non-normal (i.e. it follows "*t*" and "*F*" distribution) respectively.

To highlight the performance of the studied estimators in Tables 1–6, we use **boldface** to indicate the more efficient estimator. Proposed *BQR* estimators have resulted in reduced *MSE* as compared to baseline estimators in almost every considered scenario. Hence, the results so far are encouraging. Another key finding is that a substantial reduction in *MSE* can be noted while using the *BQR* estimator when $\rho = 0.99$ which is evident from Figs 1–5. For instance, there is an 87%, 94%, and 91% reduction in *MSE* of *HK* when $\rho = 0.99$ and $\sigma^2 = 0.5, 1, 5$ respectively in cases where $n = 25$, $p = 4$ and error term follows a normal distribution (see Fig 1). Also, the reduction in *MSE* of *HK* for the same level of collinearity and error variances are 74%, 76%, and 81%, when $n = 50$ and $p = 8$ (see Fig 2). The results are an indication that bootstrapping can serve as an instrument for boosting the efficiency of already existing ridge estimators.

When the probability distribution of the error term is *t(2)* or *F(4,16)*, a similar improved performance of the *BQR* can be noted as it exhibits in case of normal errors (see Tables 4 and 5, Figs 4 and 5). Here, the proposed *BQR* version of *AM* i.e. $AM_\gamma^*$ estimator comes out to be the most efficient as it exhibits a maximum reduction in *MSE* in almost every considered scenario. In the case where the error term is normally distributed *BQR* can reduce the *MSE* of *AM* by up to 95%. Whereas the maximum reduction in *MSE* of estimator *HK* is noted to be 93% and 97% when the error term follows a *t*-distribution and *F*-distribution respectively.

**Table 4. Estimated *MSE* when the distribution of error term is standardized *t*–distribution with 2 degrees of freedom.**

| $\rho$ | OLS | HK | HK$_\gamma^*$ | HKB | HKB$_\gamma^*$ | HSL | HSL$_\gamma^*$ | AM | AM$_\gamma^*$ | GM | GM$_\gamma^*$ |
|---|---|---|---|---|---|---|---|---|---|---|---|
| | | | | | $n = 25, p = 4$ | | | | | | |
| 0.70 | 3.857 | 1.824 | **0.540** | 1.693 | **0.819** | 2.046 | **1.281** | 0.596 | **0.408** | 0.673 | **0.408** |
| 0.80 | 15.432 | 2.757 | **0.718** | 2.164 | **1.038** | 2.355 | **1.418** | 0.613 | **0.319** | 0.959 | **0.433** |
| 0.90 | 18.291 | 4.829 | **1.313** | 3.985 | **1.326** | 3.449 | **1.757** | 0.799 | **0.260** | 1.290 | **0.359** |
| 0.99 | 150.882 | 39.18 | **2.690** | 60.732 | **8.300** | 11.024 | **0.854** | 1.150 | **0.249** | 5.233 | **0.693** |
| | | | | | $n = 50, p = 8$ | | | | | | |
| 0.70 | 7.046 | 2.623 | **1.284** | 1.656 | **1.236** | 2.639 | **2.345** | 0.474 | **0.281** | 1.616 | **0.739** |
| 0.80 | 16.889 | 3.904 | **1.780** | 2.179 | **1.630** | 3.243 | **2.847** | 0.493 | **0.208** | 0.589 | **0.249** |
| 0.90 | 20.315 | 8.095 | **3.020** | 4.979 | **3.228** | 5.155 | **3.223** | 0.471 | **0.199** | 0.939 | **0.291** |
| 0.99 | 160.586 | 62.558 | **9.840** | 38.215 | **21.482** | 14.575 | **1.593** | 1.211 | **0.112** | 6.356 | **1.843** |
| | | | | | $n = 100, p = 10$ | | | | | | |
| 0.70 | 3.632 | 1.884 | **1.194** | 1.008 | **0.799** | 1.643 | **1.537** | 0.481 | **0.169** | 0.419 | **0.211** |
| 0.80 | 5.110 | 3.165 | **1.682** | 1.684 | **1.278** | 2.349 | **1.982** | 0.405 | **0.124** | 0.355 | **0.153** |
| 0.90 | 18.74 | 5.319 | **2.833** | 3.204 | **2.397** | 4.094 | **3.528** | 0.349 | **0.099** | 0.640 | **0.242** |
| 0.99 | 160.1 | 40.28 | **11.81** | 23.73 | **15.36** | 3.446 | **1.496** | 0.709 | **0.064** | 4.332 | **1.490** |

Moreover, the *MSE* tends to increase with an increase in error variance. It is also noted that increase in the degree of collinearity generally increases the *MSE* of all estimators, however in the case of severe collinearity i.e. $\rho = 0.99$ and $\sigma^2 = 1$, some estimators show a decrease in the estimated value of *MSE* e.g. when $n = 25$ and $p = 4$ decrease in *MSE* value of the estimators such as $HK_\gamma^*$, $HKB_\gamma^*$, $HKL$, $HSL_\gamma^*$, $AM_\gamma^*$ can be observed (see Table 1). Similarly when $n = 50$ and $p = 4$ the *MSE* value of the estimator $AM_\gamma^*$ decreases (see Table 2), also when $n = 100$ and $p = 10$ the value *MSE* of the estimator $AM$, $AM_\gamma^*$, $GM_\gamma^*$ decreases (see Table 3).

Nevertheless, it is to be noticed that the *MSE* of *BQR* estimators are appreciably lower in almost all the considered simulations scenarios. Despite, increase in sample size or number of predictors. The results in Table 6 shows similar excellent performance of *BQR* estimators as

**Table 5. Estimated *MSE* when the distribution of error term is standardized *F*(4,16) distribution.**

| $\rho$ | OLS | HK | HK$_\gamma^*$ | HKB | HKB$_\gamma^*$ | HSL | HSL$_\gamma^*$ | AM | AM$_\gamma^*$ | GM | GM$_\gamma^*$ |
|---|---|---|---|---|---|---|---|---|---|---|---|
| | | | | | $n = 25, p = 4$ | | | | | | |
| 0.70 | 0.498 | 0.188 | **0.165** | 0.130 | **0.119** | 0.196 | **0.146** | 0.485 | **0.119** | 0.215 | **0.112** |
| 0.80 | 0.752 | 0.269 | **0.123** | 0.175 | **0.107** | 0.300 | **0.154** | 0.468 | **0.123** | 0.199 | **0.109** |
| 0.90 | 2.607 | 0.333 | **0.110** | 0.194 | **0.099** | 0.274 | **0.122** | 0.347 | **0.109** | 0.151 | **0.085** |
| 0.99 | 21.481 | 3.351 | **0.095** | 2.078 | **0.521** | 0.171 | **0.054** | 0.268 | **0.054** | 0.529 | **0.054** |
| | | | | | $n = 50, p = 8$ | | | | | | |
| 0.70 | 0.406 | 0.305 | **0.198** | 0.127 | **0.083** | 0.311 | **0.234** | 0.553 | **0.072** | 0.137 | **0.067** |
| 0.80 | 0.677 | 0.366 | **0.228** | 0.147 | **0.091** | 0.356 | **0.255** | 0.518 | **0.070** | 0.123 | **0.066** |
| 0.90 | 1.467 | 0.531 | **0.269** | 0.211 | **0.115** | 0.463 | **0.308** | 0.333 | **0.047** | 0.081 | **0.044** |
| 0.99 | 16.212 | 3.199 | **0.448** | 1.421 | **0.502** | 0.486 | **0.160** | 0.182 | **0.019** | 0.417 | **0.071** |
| | | | | | $n = 100, p = 10$ | | | | | | |
| 0.70 | 0.288 | 0.222 | **0.190** | 0.100 | **0.081** | 0.223 | **0.201** | 0.580 | **0.065** | 0.112 | **0.047** |
| 0.80 | 0.364 | 0.257 | **0.214** | 0.106 | **0.082** | 0.258 | **0.224** | 0.503 | **0.047** | 0.087 | **0.039** |
| 0.90 | 1.072 | 0.505 | **0.320** | 0.184 | **0.111** | 0.471 | **0.346** | 0.330 | **0.026** | 0.045 | **0.025** |
| 0.99 | 9.553 | 2.469 | **0.687** | 0.984 | **0.456** | 0.805 | **0.378** | 0.138 | **0.018** | 0.189 | **0.043** |

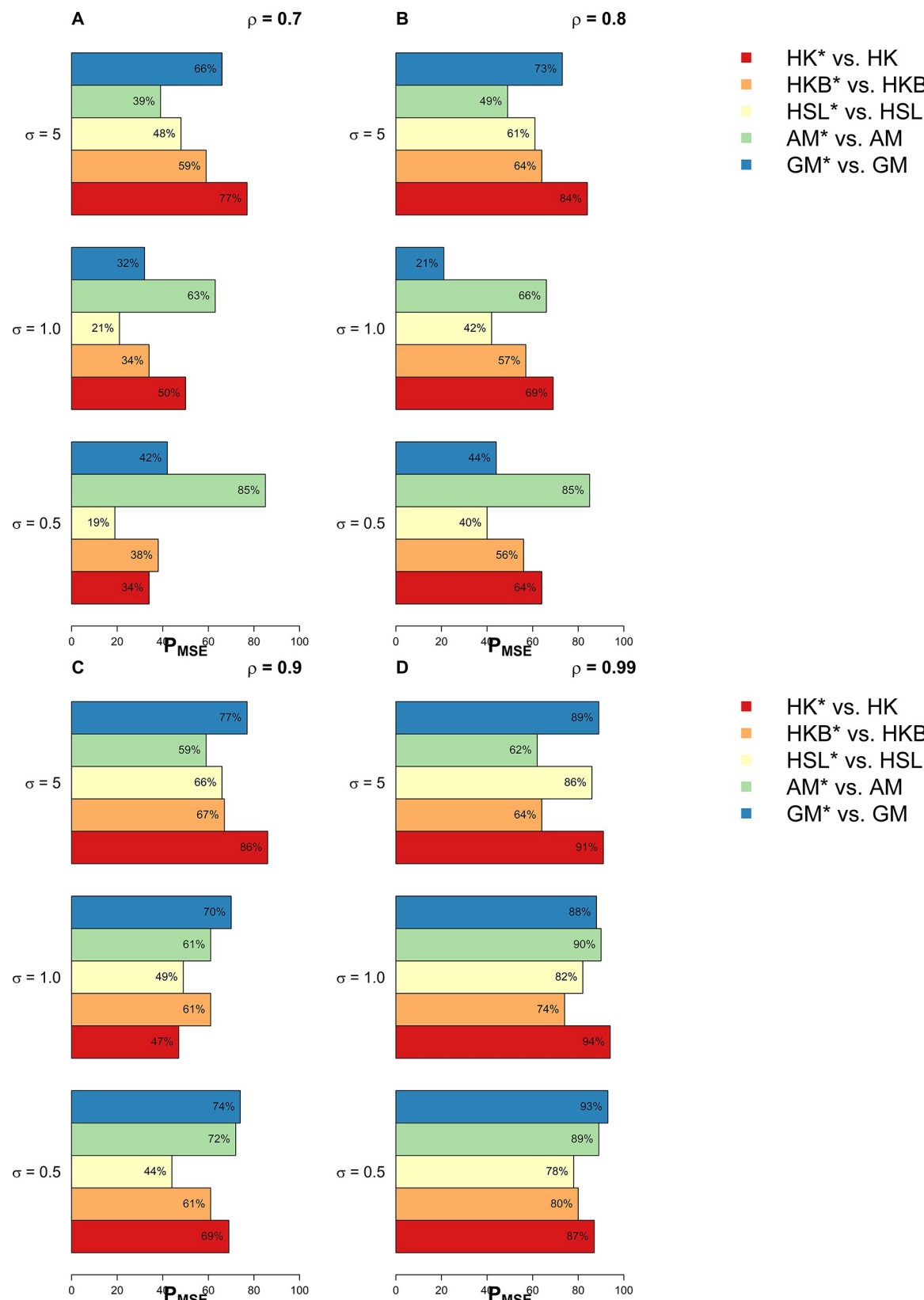

**Fig 1. Improvements by proposed ridge estimators with different values of $\rho$ when $n = 25$ and $p = 4$.**

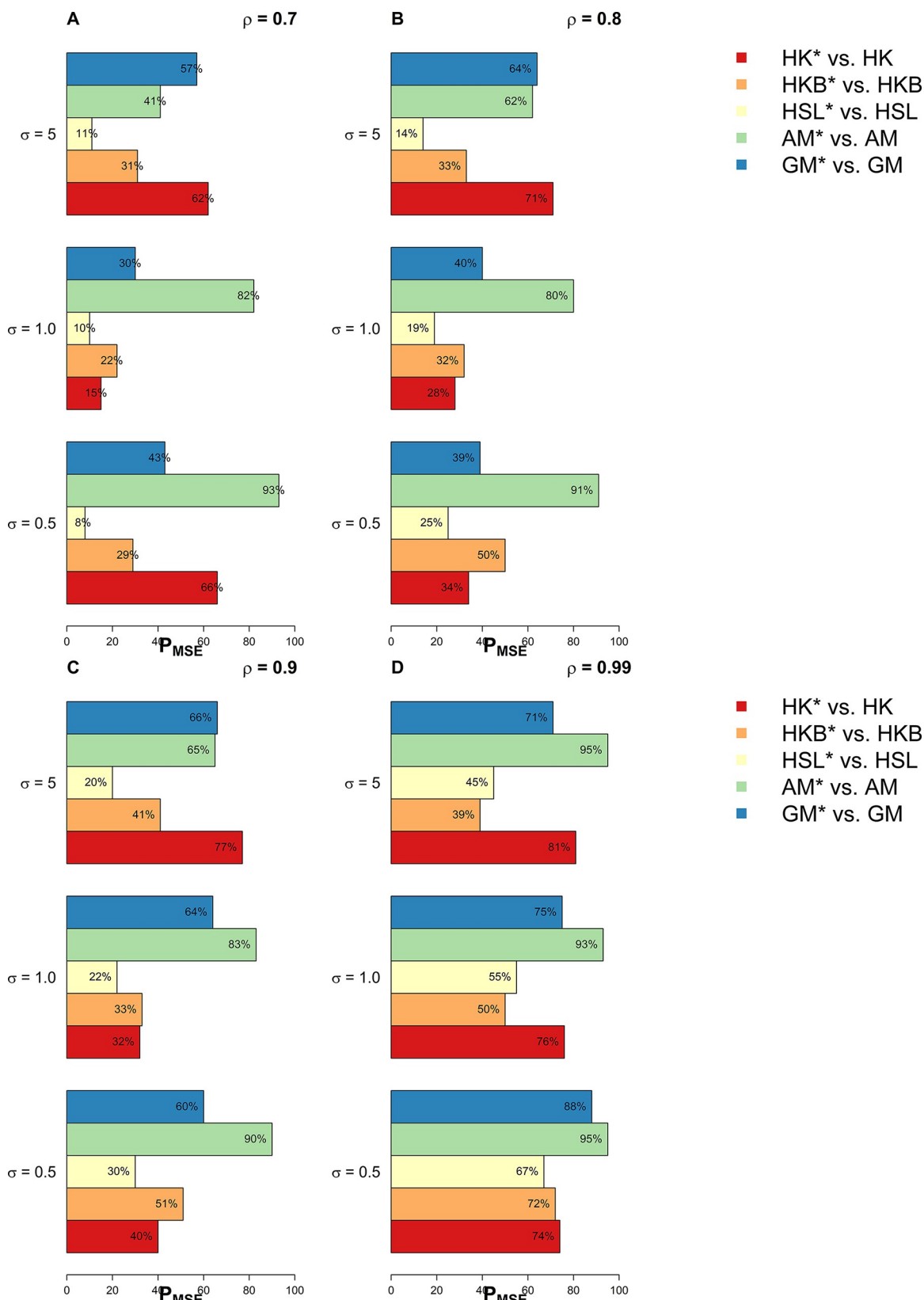

**Fig 2. Improvements by proposed ridge estimators with different values of $\rho$ when $n = 50$ and $p = 8$.**

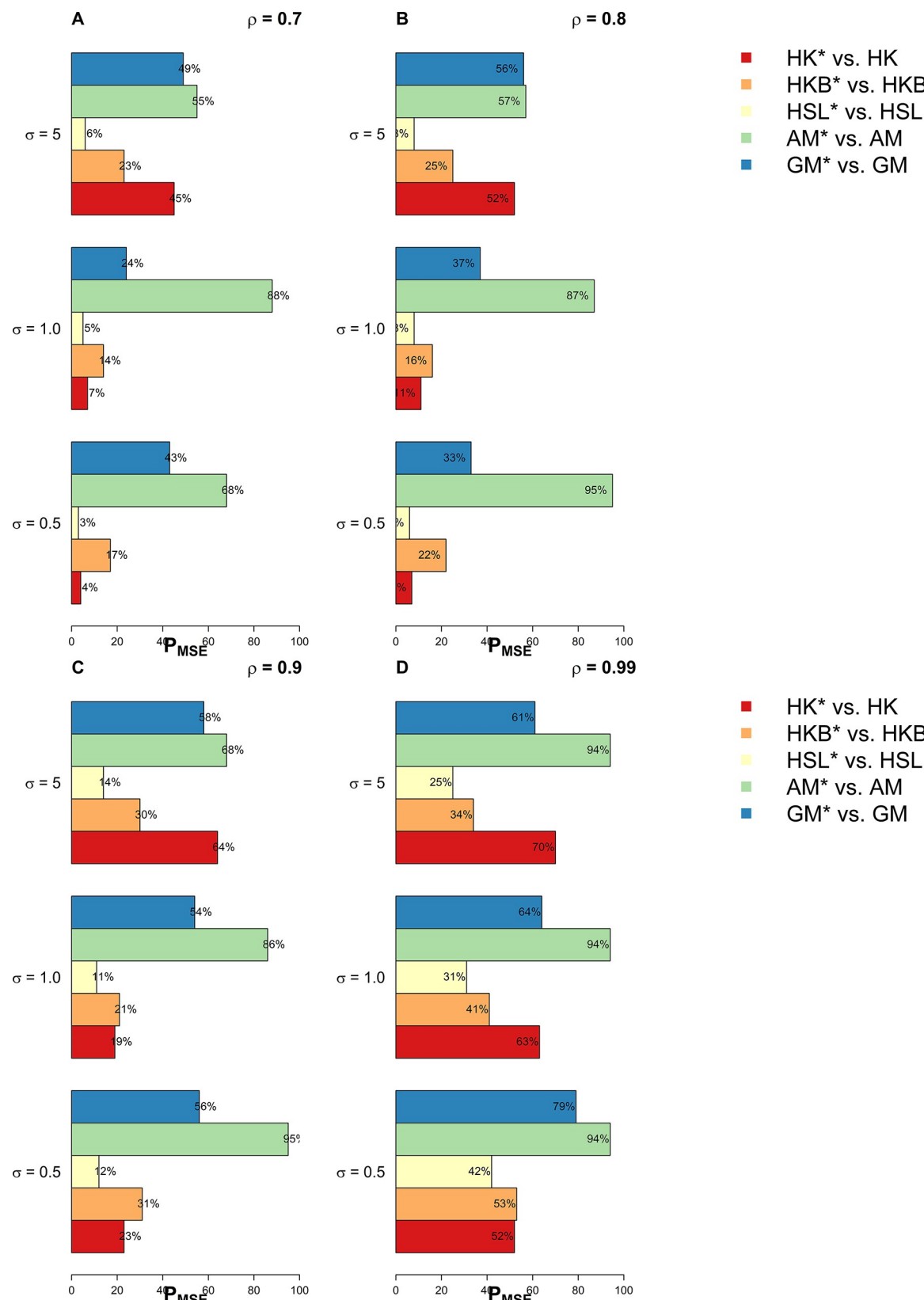

**Fig 3. Improvements by proposed ridge estimators with different values of $\rho$ when $n = 100$ and $p = 10$.**

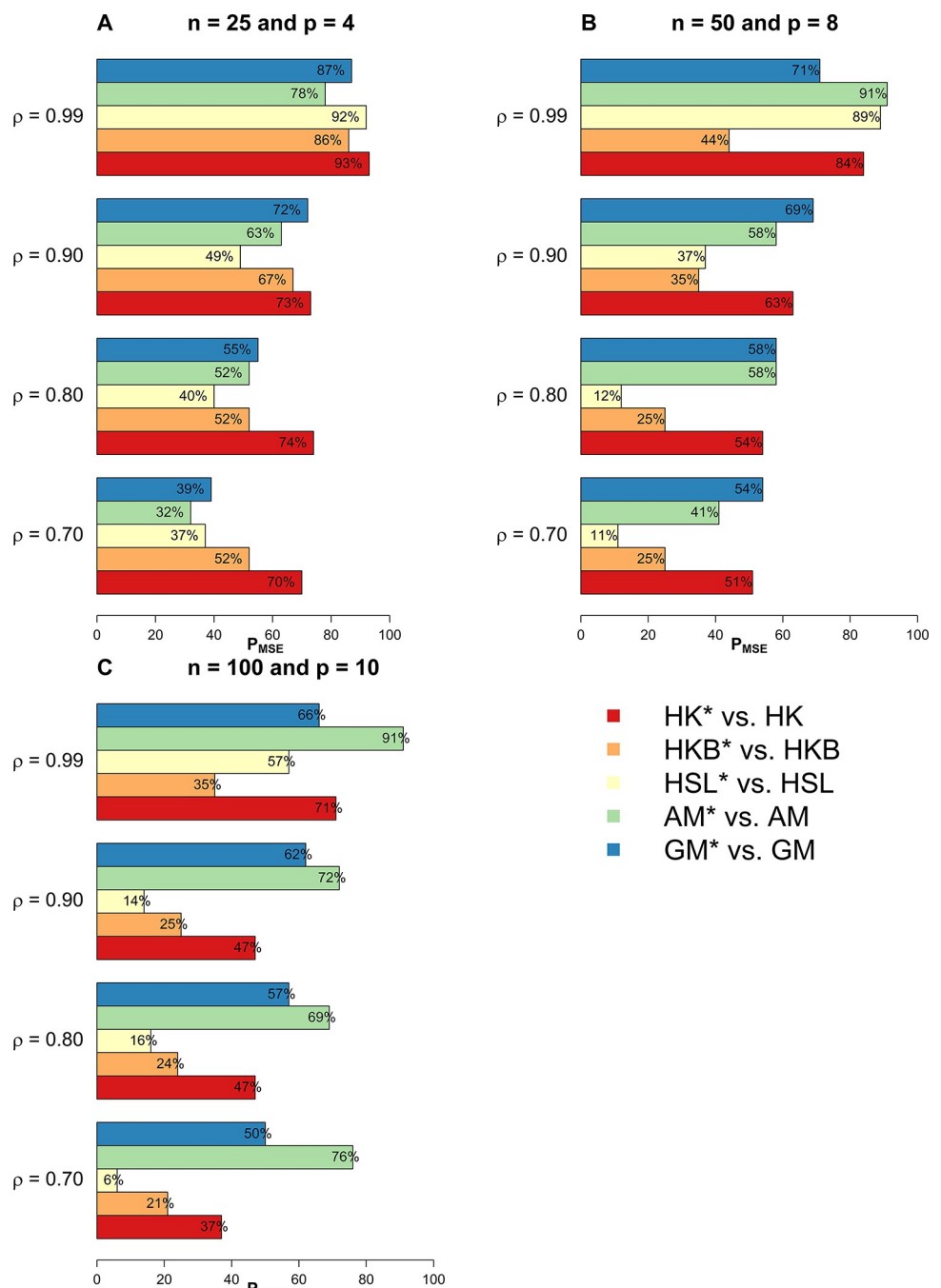

**Fig 4. Improvements by proposed ridge estimators when error term follows t distribution.**

observed in Tables 1–3 in terms of minimum *MSE*, suggesting the superiority of suggested estimators over their counterparts even when the sample size is large.

## 4. Real-life application

In this section, to illustrate the use of our proposed estimators and methodology, we have considered two real-life published datasets Tobacco data and Hospital manpower data set [39].

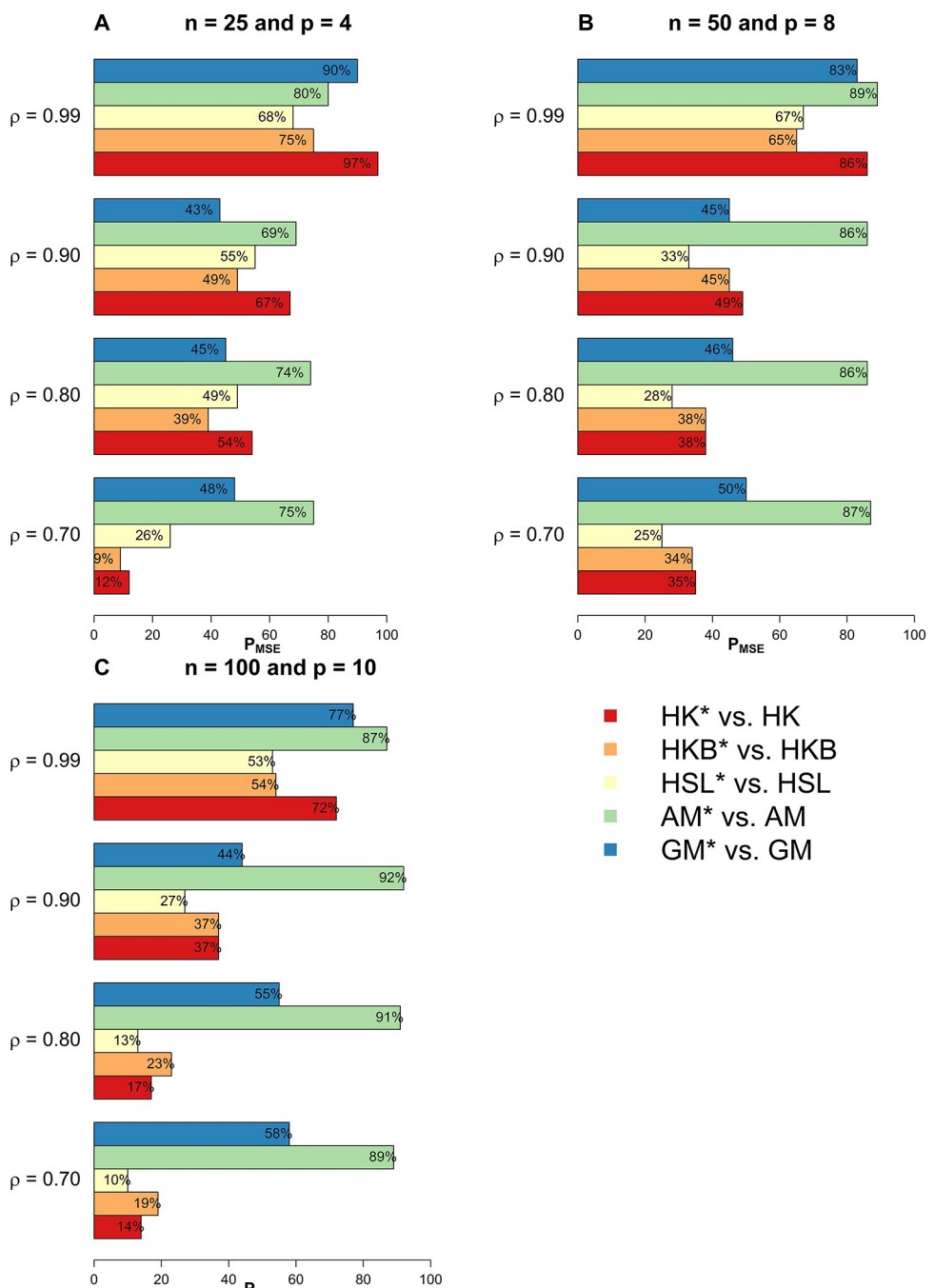

**Fig 5. Improvements by proposed ridge estimators when error term follows F distribution.**

These datasets are generally aligned with the structure we used earlier in our simulation work in Section 3.1.

## 4.1 Tobacco data

This data set consists of 30 observations; the dependent variable is the heat evolved from tobacco during the smoking process with percentage concentration of four important

**Table 6. Estimated *MSE* when the distribution of error term is $N(0,\sigma^2)$ with $n = 200, \sigma^2 = 1$ and $\rho = 0.99$. The values in parenthesis indicate reduction (%) in *MSE* due to novel *BQR* method as compared to the baseline counterpart.**

| p | OLS | HK | $HK_\gamma^*$ | HKB | $HKB_\gamma^*$ | HSL | $HSL_\gamma^*$ | AM | $AM_\gamma^*$ | GM | $GM_\gamma^*$ |
|---|-----|-----|------|------|------|------|------|------|------|------|------|
| 4 | 1.548 | 0.718 | **0.244** (66%) | 0.553 | **0.363** (34%) | 0.541 | **0.450** (17%) | 0.499 | **0.047** (91%) | 0.232 | **0.096** (58%) |
| 8 | 3.396 | 1.537 | **1.014** (34%) | 1.406 | **0.604** (57%) | 1.178 | **1.000** (15%) | 0.145 | **0.008** (94%) | 0.257 | **0.098** (61%) |
| 10 | 3.853 | 1.857 | **1.221** (34%) | 0.977 | **0.687** (30%) | 1.378 | **1.191** (14%) | 0.126 | **0.009** (93%) | 0.278 | **0.105** (62%) |
| 16 | 7.879 | 3.784 | **2.268** (40%) | 1.740 | **1.360** (22%) | 2.417 | **2.048** (15%) | 0.128 | **0.008** (94%) | 0.405 | **0.326** (20%) |
| 32 | 18.879 | 10.114 | **6.425** (36%) | 4.043 | **3.479** (14%) | 5.522 | **4.045** (27%) | 0.109 | **0.007** (94%) | 0.665 | **0.508** (24%) |

components that are taken as independent variables. The linear regression model is given as:

$$y = \beta_0 + \beta_1 X_1 + \beta_2 X_2 + \beta_3 X_3 + \beta_4 X_4 + \varepsilon_j.$$

The condition number (CN) which is the ratio of maximum eigen value and minimum eigen value is calculated as 1855.526, and also the variance inflation factor (VIF) of all

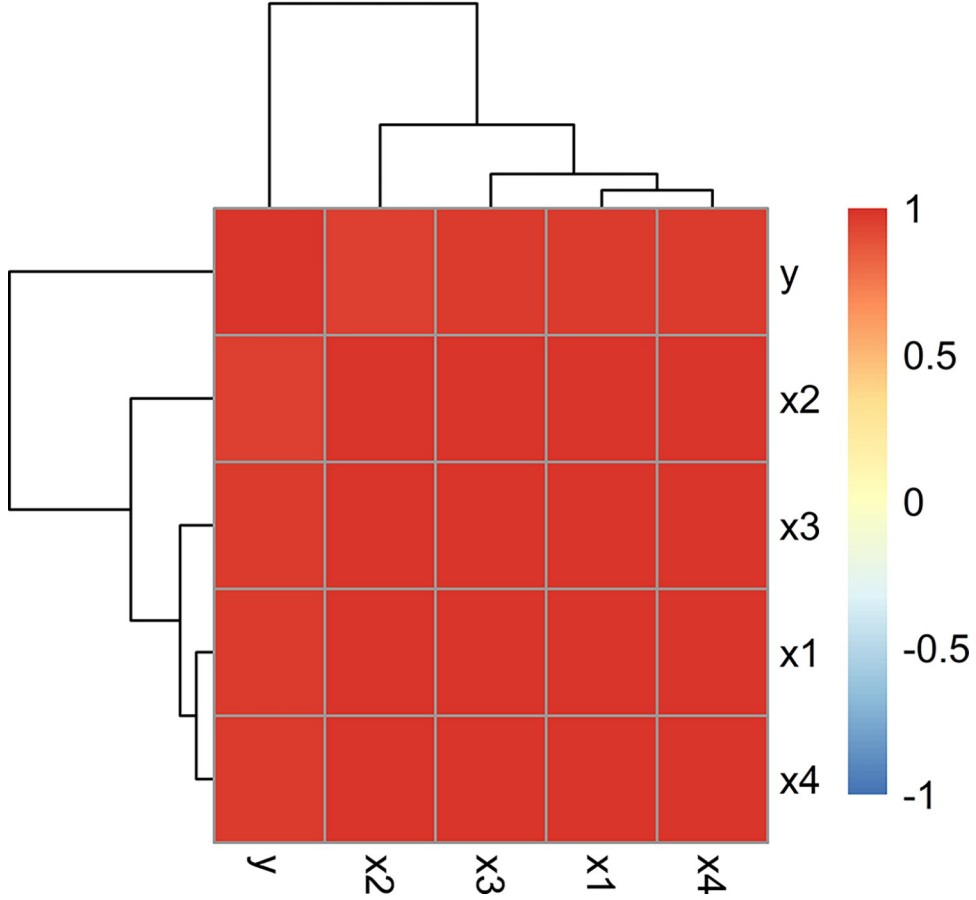

**Fig 6. Graphical display of a correlation between variables of Tobacco data.**

**Table 7. Estimated *MSEs* and estimated regression coefficients of the Tobacco data.**

| Estimator | k | MSE | $\hat{\beta}_1$ | $\hat{\beta}_2$ | $\hat{\beta}_3$ | $\hat{\beta}_4$ |
|---|---|---|---|---|---|---|
| OLS | 0 | 1.1206 | 1.5074 | -0.5211 | -0.8416 | 0.8217 |
| HK | 0.02 | 0.7749 | 1.2083 | -0.4658 | -0.611 | 0.8354 |
| $HK_{\hat{\gamma}}^*$ | 0.08 | 0.6270 | 0.8344 | -0.3613 | -0.2856 | 0.7799 |
| HKB | 0.05 | 0.6449 | 0.9877 | -0.4116 | -0.4254 | 0.8165 |
| $HKB_{\hat{\gamma}}^*$ | 0.06 | 0.6365 | 0.9583 | -0.4030 | -0.3994 | 0.8113 |
| HSL | 0.02 | 0.7753 | 1.2089 | -0.4660 | -0.6114 | 0.8354 |
| $HSL_{\hat{\gamma}}^*$ | 0.11 | 0.6510 | 0.7422 | -0.3227 | -0.1963 | 0.7442 |
| AM | 0.08 | 0.6279 | 0.8271 | -0.3585 | -0.2787 | 0.7775 |
| $AM_{\hat{\gamma}}^*$ | **0.07** | **0.6254** | **0.8816** | **-0.3783** | **-0.3297** | **0.7938** |
| GM | 0.06 | 0.6277 | 0.903 | -0.3856 | -0.3494 | 0.7993 |
| $GM_{\hat{\gamma}}^*$ | 0.06 | 0.6268 | 0.9109 | -0.3882 | -0.3566 | 0.8012 |

Note: Bold value represents the estimator with the smallest *MSE*.

predictors is greater than 10 i.e. for $X_1$, $X_2$, $X_3$ and $X_4$ the VIF calculated are 324.141, 45.173, 173.258 and 138.175. This is an indication of the presence of severe multicollinearity in the data. Fig 6 represents correlation among the variables of Tobacco data. Table 7 provides the estimated *MSE* values and regression coefficients for the proposed as well as the baseline estimator.

It is evident from Table 7 that all the *BQR* estimators outperform *OLS* as well as their baseline estimators in terms of smallest *MSE*.

## 4.2 Hospital manpower data

The data consists of 17 observations and 5 explanatory variables such as *Load* (monthly man hours), *Xray* (monthly X-ray exposures), *BedDays* (monthly occupied bed days), *AreaPop* (eligible population in the area in thousands) and *Stay* (average length of patient's stay in days). Whereas dependent variable is Hours (average daily patient load).

The linear regression model is given as:

$$Hour = \beta_0 + \beta_1 Load + \beta_2 Xray + \beta_3 BedDays + \beta_4 AreaPop + \beta_5 Stay + \varepsilon_j.$$

Fig 7 represents correlation among the variables of Hospital manpower data. The CN is calculated as. 77769.66. Also, the VIF of the predictors *Loads*, *BedDays* and *AreaPop* are 9597.57, 8933.09 and 23.29 which are greater than 10; these numbers indicate a strong multicollinearity problem among the predictors. The estimated *MSE* for each estimator of manpower data are given in Table 8. From this table, it is evident that proposed *BQR* estimators have smaller *MSE* as compared to their base line estimator. This suggests that the proposed estimators outperform their competitors, suggesting the superiority of these estimators over the baseline estimators.

## 5. Conclusion

In this article, the bootstrap-quantile approach is suggested for the improvement of existing ridge estimators and thus the efficient estimation of regression coefficients. Since the ridge parameter adjusts the amount of shrinkage therefore its optimization is an important task to obtain better regression estimates. Using the resampling mechanism inherent in

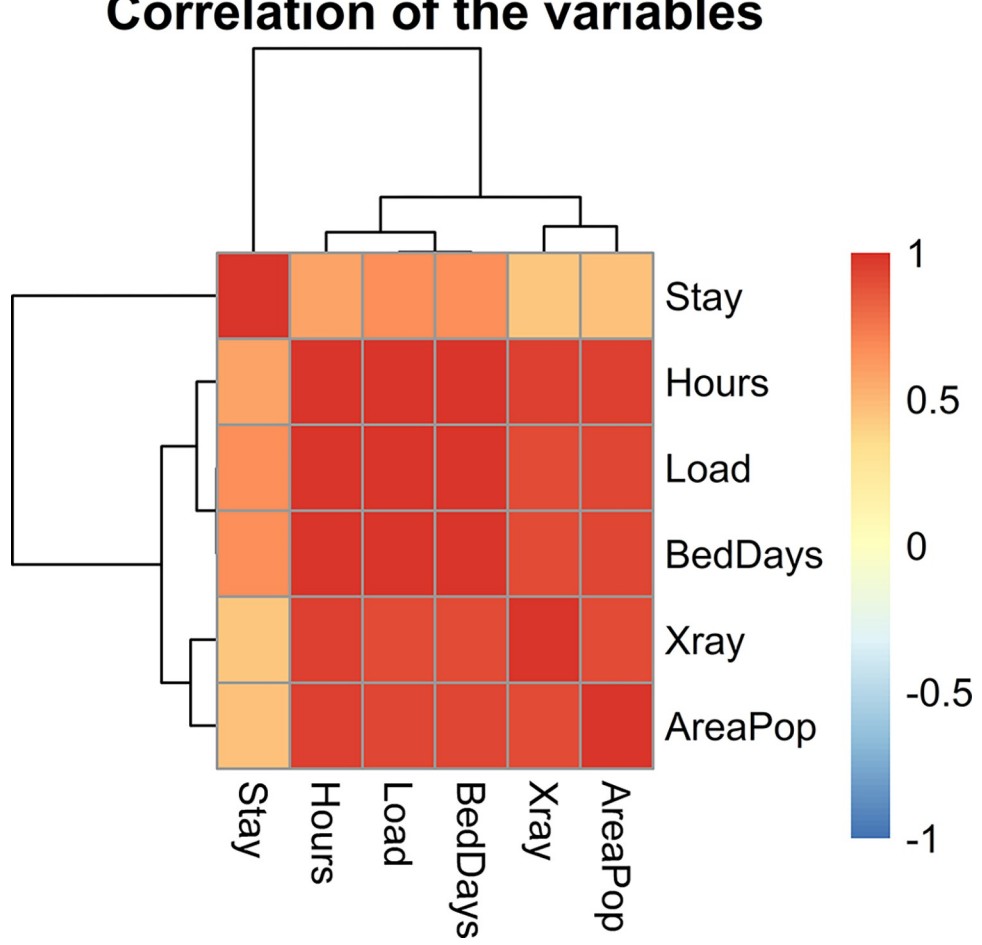

**Fig 7. Graphical display of a correlation between variables of Hospital manpower data.**

**Table 8. Estimated *MSEs* and estimated regression coefficients of the Hospital manpower data.**

| Estimator | K | MSE | $\hat{\beta}_1$ | $\hat{\beta}_2$ | $\hat{\beta}_3$ | $\hat{\beta}_4$ | $\hat{\beta}_5$ |
|---|---|---|---|---|---|---|---|
| OLS | 0 | 14.1831 | -0.4591 | 0.2140 | 1.4027 | -0.0819 | -0.1123 |
| HK | 0.01 | 0.2483 | 0.3603 | 0.2161 | 0.6073 | -0.1053 | -0.1186 |
| $HK_\gamma^*$ | **0.08** | **0.0424** | **0.4400** | **0.2231** | **0.4811** | **-0.0750** | **-0.1041** |
| HKB | 0.03 | 0.0522 | 0.4380 | 0.2185 | 0.5164 | -0.0972 | -0.1144 |
| $HKB_\gamma^*$ | 0.14 | 0.0487 | 0.4255 | 0.2285 | 0.4551 | -0.0477 | -0.0916 |
| HSL | 0.04 | 0.0461 | 0.4422 | 0.2195 | 0.5048 | -0.0923 | -0.1121 |
| $HSL_\gamma^*$ | 0.05 | 0.0437 | 0.4431 | 0.2205 | 0.4971 | -0.0877 | -0.1100 |
| AM | 0.62 | 0.1922 | 0.3519 | 0.2458 | 0.3661 | 0.0618 | -0.0402 |
| $AM_\gamma^*$ | 0.16 | 0.0519 | 0.4211 | 0.2298 | 0.4489 | -0.0405 | -0.0883 |
| GM | 0.18 | 0.0539 | 0.4187 | 0.2305 | 0.4456 | -0.0366 | -0.0865 |
| $GM_\gamma^*$ | 0.1 | 0.0438 | 0.4348 | 0.2253 | 0.4699 | -0.0639 | -0.0991 |

Note: Bold value represents the estimator with the smallest *MSE*.

bootstrapping, it is demonstrated that our proposed method, i.e. *BQR* method remarkably improves the performance of any ridge estimator. We have also studied applications to tobacco and hospital manpower data to illustrate the use of our proposed method. The bootstrap methods, especially the wild bootstrapping, can be further studied for the regression models with multicollinear predictors and heteroscedastic errors.

## Supporting information

**S1 Text.**
(TXT)

## Author Contributions

**Conceptualization:** Sohail Chand.

**Data curation:** Irum Sajjad Dar.

**Formal analysis:** Irum Sajjad Dar.

**Methodology:** Sohail Chand.

**Supervision:** Sohail Chand.

**Writing – original draft:** Irum Sajjad Dar.

**Writing – review & editing:** Sohail Chand.

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
