## [Decision Letter · Decision Letter 0]

26 Sep 2023

PONE-D-23-23113Bootstrap-quantile ridge estimator for linear regression with application to the groundwater physiochemical quality parametersPLOS ONE

Dear Dr. Chand,

Thank you for submitting your manuscript to PLOS ONE. After careful consideration, we feel that it has merit but does not fully meet PLOS ONE’s publication criteria as it currently stands. Therefore, we invite you to submit a revised version of the manuscript that addresses the points raised during the review process.

We look forward to receiving your revised manuscript.

Kind regards,

Md Hasinur Rahaman Khan, Ph.D.

Academic Editor

PLOS ONE

Additional Editor Comments:

In addition to the reviewer's comment, I have some points that the authors need to address:

1. Bootsrap generally considers two-third observations and hence produces underestimated results. Have you mentioned this as one of the demerits of your proposal.

2. Ridge-estimator is not used if there is correlated covariates in the model. Instead elastic-net is generally preferred for overcoming this issue. What is the rational of using Ridge estimator?

3. May need to cite some related papers:

M.H.R. Khan, A. Bhadra and T. Howlader (2019). Stability Selection for Lasso, Ridge and Elastic Net Implemented with AFT Models. Statistical Applications in Genetics and Molecular Biology, 18(5), 20170001

M.H.R. Khan and J.E.H. Shaw (2016). Variable Selection for Survival Data with a Class of Adaptive Elastic Net Techniques. Statistics and Computing, 26(3): 725-741

M.H.R. Khan (2018). On The Performance of Adaptive Pre-processing Technique in Analysing High-dimensional Censored Data. Biometrical Journal, 60(4): 687-702

Reviewers' comments:

Reviewer's Responses to Questions

**Comments to the Author**

1. Is the manuscript technically sound, and do the data support the conclusions?

Reviewer #1: Yes

Reviewer #2: Partly

2. Has the statistical analysis been performed appropriately and rigorously? 

Reviewer #1: Yes

Reviewer #2: No

3. Have the authors made all data underlying the findings in their manuscript fully available?

Reviewer #1: Yes

Reviewer #2: Yes

4. Is the manuscript presented in an intelligible fashion and written in standard English?

Reviewer #1: No

Reviewer #2: No

5. Review Comments to the Author

Reviewer #1: In my opinion, the paper offers a good contribution. So, I recommend accepting this paper, but after making the following major modifications:

1. I think that some recent papers related to "the ridge estimator" should be mentioned such as:

- DOI: 10.3390/stats3040033; DOI:10.3389/fams.2022.952142; DOI:10.1080/03610918.2021.1960373; DOI: 10.1016/j.sciaf.2022.e01372; DOI: 10.1080/03610918.2021.1939374.

2. In the simulation study, the author should use more different sample sizes, such as n>100. Moreover, it is very important to display the 'bias' amount of different biased estimators in the simulation study.

3. In application (table 8), what are the values of k's used for?

4. The author should put "datasets" that are used in the section of applications in a "Supporting Information File".

5. In section of conclusion, add a sentence about future work.

6. There are some grammatical errors in the paper. The author needs to carefully review the full text.

Reviewer #2: This paper develops a nonparametric bootstrap-quantile approach for the estimation of ridge parameter in linear regression model. The proposed method is illustrated using some popular and widely used ridge estimators but this idea can be extended to any ridge estimator. Monte Carlo simulations are carried out to compare the performance of the proposed estimators with their baseline counterparts. It is demonstrated empirically that MSE obtained from

our suggested bootstrap-quantile approach are substantially smaller than their baseline estimators especially when collinearity is high. Application to physiochemical properties of water quality data reveals the suitability of the idea.

The novelty of the article must be highlighted in relation to the existing literature. Many articles related to this topic has been published in the literature so what is new?

Why MSE is used to assess the performance?

What is the theoretical justification of the proposed method. It is suggested to the authors to derive the properties of the proposed estimators, especially large sample properties.

How the initial values are selected?

A new real data set should be used.

How the initial values are taken to initialized the algorithm in real data case?

Literature review must be updated by including all recent contribution in ridge regression.

The complete computational code should be submitted to verify the reproducibility.

6. PLOS authors have the option to publish the peer review history of their article (what does this mean?). If published, this will include your full peer review and any attached files.

Reviewer #1: No

Reviewer #2: No

---

## [Decision Letter · Decision Letter 1]

8 Mar 2024

PONE-D-23-23113R1Bootstrap-quantile ridge estimator for linear regression with applicationsPLOS ONE

Dear Dr. Chand,

Thank you for submitting your manuscript to PLOS ONE. After careful consideration, we feel that it has merit but does not fully meet PLOS ONE’s publication criteria as it currently stands. Therefore, we invite you to submit a revised version of the manuscript that addresses the points raised during the review process.

We look forward to receiving your revised manuscript.

Kind regards,

Mohamed R. Abonazel, Ph.D.

Academic Editor

PLOS ONE

Reviewers' comments:

Reviewer's Responses to Questions

**Comments to the Author**

1. If the authors have adequately addressed your comments raised in a previous round of review and you feel that this manuscript is now acceptable for publication, you may indicate that here to bypass the “Comments to the Author” section, enter your conflict of interest statement in the “Confidential to Editor” section, and submit your "Accept" recommendation.

Reviewer #2: (No Response)

2. Is the manuscript technically sound, and do the data support the conclusions?

Reviewer #2: Partly

3. Has the statistical analysis been performed appropriately and rigorously? 

Reviewer #2: Yes

4. Have the authors made all data underlying the findings in their manuscript fully available?

Reviewer #2: No

5. Is the manuscript presented in an intelligible fashion and written in standard English?

Reviewer #2: Yes

6. Review Comments to the Author

Reviewer #2: This paper develops a nonparametric bootstrap-quantile approach for the estimation of ridge parameter in linear regression model. The proposed method is illustrated using some popular and widely used ridge estimators but this idea can be extended to any ridge estimator. Monte Carlo simulations are carried out to compare the performance of the proposed estimators with their baseline counterparts. It is demonstrated empirically that MSE obtained from

our suggested bootstrap-quantile approach are substantially smaller than their baseline estimators especially when collinearity is high. Application to physiochemical properties of water quality data reveals the suitability of the idea.

The authors tried to address previous comments but not fully. For example,

There is no attachment showing the complete computational code to verify the reproducibility.

The source of 'Hospital manpower data' is not mentioned.

Why is not MAPE or MAE used rather than MSE?

The theoretical justification of bootstrap is available in books. Please cite an appropriate reference.

Can we consider a situation of n=10, and p=15?

7. PLOS authors have the option to publish the peer review history of their article (what does this mean?). If published, this will include your full peer review and any attached files.

Reviewer #2: No

---

## [Author Response · Author response to Decision Letter 1]

20 Mar 2024

We are thankful to the Editor and Reviewers for their comments. We have revised the draft according to their suggestions.

---

## [Decision Letter · Decision Letter 2]

1 Apr 2024

Bootstrap-quantile ridge estimator for linear regression with applications

PONE-D-23-23113R2

Dear Dr. Chand,

We’re pleased to inform you that your manuscript has been judged scientifically suitable for publication and will be formally accepted for publication once it meets all outstanding technical requirements.

Kind regards,

Mohamed R. Abonazel, Ph.D.

Academic Editor

PLOS ONE

Additional Editor Comments (optional):

Reviewers' comments:

Reviewer's Responses to Questions

**Comments to the Author**

1. If the authors have adequately addressed your comments raised in a previous round of review and you feel that this manuscript is now acceptable for publication, you may indicate that here to bypass the “Comments to the Author” section, enter your conflict of interest statement in the “Confidential to Editor” section, and submit your "Accept" recommendation.

Reviewer #2: All comments have been addressed

2. Is the manuscript technically sound, and do the data support the conclusions?

Reviewer #2: Yes

3. Has the statistical analysis been performed appropriately and rigorously? 

Reviewer #2: Yes

4. Have the authors made all data underlying the findings in their manuscript fully available?

Reviewer #2: Yes

5. Is the manuscript presented in an intelligible fashion and written in standard English?

Reviewer #2: Yes

6. Review Comments to the Author

Reviewer #2: This paper develops a nonparametric bootstrap-quantile approach for the estimation of ridge parameter in linear regression model. The proposed method is illustrated using some popular and widely used ridge estimators but this idea can be extended to any ridge estimator. Monte Carlo simulations are carried out to compare the performance of the proposed estimators with their baseline counterparts. It is demonstrated empirically that MSE obtained from

our suggested bootstrap-quantile approach are substantially smaller than their baseline estimators especially when collinearity is high. Application to physiochemical properties of water quality data reveals the suitability of the idea.

The authors addressed all comments.

7. PLOS authors have the option to publish the peer review history of their article (what does this mean?). If published, this will include your full peer review and any attached files.

Reviewer #2: No
